# Consistent Binary Classification with Generalized Performance Metrics

**Oluwasanmi Koyejo**[*]
Department of Psychology,
Stanford University
sanmi@stanford.edu

**Nagarajan Natarajan**[*]
Department of Computer Science,
University of Texas at Austin
naga86@cs.utexas.edu

**Pradeep Ravikumar**
Department of Computer Science,
University of Texas at Austin
pradeepr@cs.utexas.edu

**Inderjit S. Dhillon**
Department of Computer Science,
University of Texas at Austin
inderjit@cs.utexas.edu

## Abstract

Performance metrics for binary classification are designed to capture tradeoffs between four fundamental population quantities: true positives, false positives, true negatives and false negatives. Despite significant interest from theoretical and applied communities, little is known about either optimal classifiers or consistent algorithms for optimizing binary classification performance metrics beyond a few special cases. We consider a fairly large family of performance metrics given by ratios of linear combinations of the four fundamental population quantities. This family includes many well known binary classification metrics such as classification accuracy, AM measure, F-measure and the Jaccard similarity coefficient as special cases. Our analysis identifies the optimal classifiers as the sign of the thresholded conditional probability of the positive class, with a performance metric-dependent threshold. The optimal threshold can be constructed using simple plug-in estimators when the performance metric is a linear combination of the population quantities, but alternative techniques are required for the general case. We propose two algorithms for estimating the optimal classifiers, and prove their statistical consistency. Both algorithms are straightforward modifications of standard approaches to address the key challenge of optimal threshold selection, thus are simple to implement in practice. The first algorithm combines a plug-in estimate of the conditional probability of the positive class with optimal threshold selection. The second algorithm leverages recent work on calibrated asymmetric surrogate losses to construct candidate classifiers. We present empirical comparisons between these algorithms on benchmark datasets.

## 1 Introduction

Binary classification performance is often measured using metrics designed to address the shortcomings of classification accuracy. For instance, it is well known that classification accuracy is an inappropriate metric for rare event classification problems such as medical diagnosis, fraud detection, click rate prediction and text retrieval applications [1, 2, 3, 4]. Instead, alternative metrics better tuned to imbalanced classification (such as the $F_1$ measure) are employed. Similarly, cost-sensitive metrics may useful for addressing asymmetry in real-world costs associated with specific classes. An important theoretical question concerning metrics employed in binary classification is the characteri-

---

[*]Equal contribution to the work.

zation of the optimal decision functions. For example, the decision function that maximizes the accuracy metric (or equivalently minimizes the "0-1 loss") is well-known to be $\text{sign}(P(Y = 1|x) - 1/2)$. A similar result holds for cost-sensitive classification [5]. Recently, [6] showed that the optimal decision function for the $F_1$ measure, can also be characterized as $\text{sign}(P(Y = 1|x) - \delta^*)$ for some $\delta^* \in (0, 1)$. As we show in the paper, it is not a coincidence that the optimal decision function for these different metrics has a similar simple characterization. We make the observation that the different metrics used in practice belong to a fairly general family of performance metrics given by ratios of linear combinations of the four population quantities associated with the confusion matrix.

We consider a family of performance metrics given by ratios of linear combinations of the four population quantities. Measures in this family include classification accuracy, false positive rate, false discovery rate, precision, the AM measure and the F-measure, among others. Our analysis shows that the optimal classifiers for all such metrics can be characterized as the sign of the thresholded conditional probability of the positive class, with a threshold that depends on the specific metric. This result unifies and generalizes known special cases including the AM measure analysis by Menon et al. [7], and the $F_\beta$ measure analysis by Ye et al. [6]. It is known that minimizing (convex) surrogate losses, such as the hinge and the logistic loss, provably also minimizes the underlying 0-1 loss or equivalently maximizes the classification accuracy [8]. This motivates the next question we address in the paper: can one obtain algorithms that (a) can be used in practice for maximizing metrics from our family, and (b) are *consistent* with respect to the metric? To this end, we propose two algorithms for consistent empirical estimation of decision functions. The first algorithm combines a plug-in estimate of the conditional probability of the positive class with optimal threshold selection. The second leverages the asymmetric surrogate approach of Scott [9] to construct candidate classifiers. Both algorithms are simple modifications of standard approaches that address the key challenge of optimal threshold selection. Our analysis identifies why simple heuristics such as classification using class-weighted loss functions and logistic regression with threshold search are effective practical algorithms for many generalized performance metrics, and furthermore, that when implemented correctly, such apparent heuristics are in fact asymptotically consistent.

**Related Work.** Binary classification accuracy and its cost-sensitive variants have been studied extensively. Here we highlight a few of the key results. The seminal work of [8] showed that minimizing certain surrogate loss functions enables us to control the probability of misclassification (the expected 0-1 loss). An appealing corollary of the result is that convex loss functions such as the hinge and logistic losses satisfy the surrogacy conditions, which establishes the statistical consistency of the resulting algorithms. Steinwart [10] extended this work to derive surrogates losses for other scenarios including asymmetric classification accuracy. More recently, Scott [9] characterized the optimal decision function for weighted 0-1 loss in cost-sensitive learning and extended the risk bounds of [8] to weighted surrogate loss functions. A similar result regarding the use of a threshold different than 1/2, and appropriately rebalancing the training data in cost-sensitive learning, was shown by [5]. Surrogate regret bounds for proper losses applied to class probability estimation were analyzed by Reid and Williamson [11] for differentiable loss functions. Extensions to the multi-class setting have also been studied (for example, Zhang [12] and Tewari and Bartlett [13]). Analysis of performance metrics beyond classification accuracy is limited. The optimal classifier remains unknown for many binary classification performance metrics of interest, and few results exist for identifying consistent algorithms for optimizing these metrics [7, 6, 14, 15]. Of particular relevance to our work are the AM measure maximization by Menon et al. [7], and the $F_\beta$ measure maximization by Ye et al. [6].

## 2 Generalized Performance Metrics

Let $\mathcal{X}$ be either a countable set, or a complete separable metric space equipped with the standard Borel $\sigma$-algebra of measurable sets. Let $X \in \mathcal{X}$ and $Y \in \{0, 1\}$ represent input and output random variables respectively. Further, let $\Theta$ represent the set of all classifiers $\Theta = \{\theta : \mathcal{X} \mapsto [0, 1]\}$. We assume the existence of a fixed unknown distribution $\mathbb{P}$, and data is generated as iid. samples $(X, Y) \sim \mathbb{P}$. Define the quantities: $\pi = \mathbb{P}(Y = 1)$ and $\gamma(\theta) = \mathbb{P}(\theta = 1)$.

The components of the confusion matrix are the fundamental population quantities for binary classification. They are the true positives (TP), false positives (FP), true negatives (TN) and false negatives

(FN), given by:

$$\text{TP}(\theta, \mathbb{P}) = \mathbb{P}(Y = 1, \theta = 1), \qquad \text{FP}(\theta, \mathbb{P}) = \mathbb{P}(Y = 0, \theta = 1), \qquad (1)$$
$$\text{FN}(\theta, \mathbb{P}) = \mathbb{P}(Y = 1, \theta = 0), \qquad \text{TN}(\theta, \mathbb{P}) = \mathbb{P}(Y = 0, \theta = 0).$$

These quantities may be further decomposed as:

$$\text{FP}(\theta, \mathbb{P}) = \gamma(\theta) - \text{TP}(\theta), \quad \text{FN}(\theta, \mathbb{P}) = \pi - \text{TP}(\theta), \quad \text{TN}(\theta, \mathbb{P}) = 1 - \gamma(\theta) - \pi + \text{TP}(\theta). \quad (2)$$

Let $\mathcal{L} : \Theta \times \mathbb{P} \mapsto \mathbb{R}$ be a performance metric of interest. Without loss of generality, we assume that $\mathcal{L}$ is a utility metric, so that larger values are better. The *Bayes utility* $\mathcal{L}^*$ is the optimal value of the performance metric, i.e., $\mathcal{L}^* = \sup_{\theta \in \Theta} \mathcal{L}(\theta, \mathbb{P})$. The *Bayes classifier* $\theta^*$ is the classifier that optimizes the performance metric, so $\mathcal{L}^* = \mathcal{L}(\theta^*)$, where:

$$\theta^* = \arg\max_{\theta \in \Theta} \mathcal{L}(\theta, \mathbb{P}).$$

We consider a family of classification metrics computed as the ratio of linear combinations of these fundamental population quantities (1). In particular, given constants (representing costs or weights) $\{a_{11}, a_{10}, a_{01}, a_{00}, a_0\}$ and $\{b_{11}, b_{10}, b_{01}, b_{00}, b_0\}$, we consider the measure:

$$\mathcal{L}(\theta, \mathbb{P}) = \frac{a_0 + a_{11}\text{TP} + a_{10}\text{FP} + a_{01}\text{FN} + a_{00}\text{TN}}{b_0 + b_{11}\text{TP} + b_{10}\text{FP} + b_{01}\text{FN} + b_{00}\text{TN}} \qquad (3)$$

where, for clarity, we have suppressed dependence of the population quantities on $\theta$ and $\mathbb{P}$. Examples of performance metrics in this family include the AM measure [7], the $F_\beta$ measure [6], the Jaccard similarity coefficient (JAC) [16] and Weighted Accuracy (WA):

$$\text{AM} = \frac{1}{2}\left(\frac{\text{TP}}{\pi} + \frac{\text{TN}}{1-\pi}\right) = \frac{(1-\pi)\text{TP} + \pi\text{TN}}{2\pi(1-\pi)}, \ F_\beta = \frac{(1+\beta^2)\text{TP}}{(1+\beta^2)\text{TP} + \beta^2\text{FN} + \text{FP}} = \frac{(1+\beta^2)\text{TP}}{\beta^2\pi + \gamma},$$

$$\text{JAC} = \frac{\text{TP}}{\text{TP} + \text{FN} + \text{FP}} = \frac{\text{TP}}{\pi + \text{FP}} = \frac{\text{TP}}{\gamma + \text{FN}}, \quad \text{WA} = \frac{w_1\text{TP} + w_2\text{TN}}{w_1\text{TP} + w_2\text{TN} + w_3\text{FP} + w_4\text{FN}}.$$

Note that we allow the constants to depend on $\mathbb{P}$. Other examples in this class include commonly used ratios such as the true positive rate (also known as recall) (TPR), true negative rate (TNR), precision (Prec), false negative rate (FNR) and negative predictive value (NPV):

$$\text{TPR} = \frac{\text{TP}}{\text{TP} + \text{FN}}, \ \text{TNR} = \frac{\text{TN}}{\text{FP} + \text{TN}}, \ \text{Prec} = \frac{\text{TP}}{\text{TP} + \text{FP}}, \ \text{FNR} = \frac{\text{FN}}{\text{FN} + \text{TP}}, \ \text{NPV} = \frac{\text{TN}}{\text{TN} + \text{FN}}.$$

Interested readers are referred to [17] for a list of additional metrics in this class.

By decomposing the population measures (1) using (2) we see that any performance metric in the family (3) has the equivalent representation:

$$\mathcal{L}(\theta) = \frac{c_0 + c_1\text{TP}(\theta) + c_2\gamma(\theta)}{d_0 + d_1\text{TP}(\theta) + d_2\gamma(\theta)} \qquad (4)$$

with the constants:

$$c_0 = a_{01}\pi + a_{00} - a_{00}\pi + a_0, \quad c_1 = a_{11} - a_{10} - a_{01} + a_{00}, \quad c_2 = a_{10} - a_{00} \quad \text{and}$$
$$d_0 = b_{01}\pi + b_{00} - b_{00}\pi + b_0, \quad d_1 = b_{11} - b_{10} - b_{01} + b_{00}, \quad d_2 = b_{10} - b_{00}.$$

Thus, it is clear from (4) that the family of performance metrics depends on the classifier $\theta$ only through the quantities $\text{TP}(\theta)$ and $\gamma(\theta)$.

**Optimal Classifier**

We now characterize the optimal classifier for the family of performance metrics defined in (4). Let $\nu$ represent the dominating measure on $\mathcal{X}$. For the rest of this manuscript, we make the following assumption:

**Assumption 1.** *The marginal distribution $\mathbb{P}(X)$ is absolutely continuous with respect to the dominating measure $\nu$ on $\mathcal{X}$ so there exists a density $\mu$ that satisfies $d\mathbb{P} = \mu d\nu$.*

To simplify notation, we use the standard $d\nu(x) = dx$. We also define the conditional probability $\eta_x = \mathbb{P}(Y = 1 | X = x)$. Applying Assumption 1, we can expand the terms $\text{TP}(\theta) = \int_{x \in \mathcal{X}} \eta_x \theta(x) \mu(x) dx$ and $\gamma(\theta) = \int_{x \in \mathcal{X}} \theta(x) \mu(x) dx$, so the performance metric (4) may be represented as:

$$\mathcal{L}(\theta, \mathbb{P}) = \frac{c_0 + \int_{x \in \mathcal{X}} (c_1 \eta_x + c_2) \theta(x) \mu(x) dx}{d_0 + \int_{x \in \mathcal{X}} (d_1 \eta_x + d_2) \theta(x) \mu(x)}.$$

Our first main result identifies the Bayes classifier for all utility functions in the family (3), showing that they take the form $\theta^*(x) = \text{sign}(\eta_x - \delta^*)$, where $\delta^*$ is a metric-dependent threshold, and the sign function is given by $\text{sign} : \mathbb{R} \mapsto \{0, 1\}$ as $\text{sign}(t) = 1$ if $t \geq 0$ and $\text{sign}(t) = 0$ otherwise.

**Theorem 2.** *Let $\mathbb{P}$ be a distribution on $\mathcal{X} \times [0, 1]$ that satisfies Assumption 1, and let $\mathcal{L}$ be a performance metric in the family* (3). *Given the constants $\{c_0, c_1, c_2\}$ and $\{d_0, d_1, d_2\}$, define:*

$$\delta^* = \frac{d_2 \mathcal{L}^* - c_2}{c_1 - d_1 \mathcal{L}^*}. \tag{5}$$

1. *When $c_1 > d_1 \mathcal{L}^*$, the Bayes classifier $\theta^*$ takes the form $\theta^*(x) = \text{sign}(\eta_x - \delta^*)$*

2. *When $c_1 < d_1 \mathcal{L}^*$, the Bayes classifier takes the form $\theta^*(x) = \text{sign}(\delta^* - \eta_x)$*

The proof of the theorem involves examining the first-order optimality condition (see Appendix B).

**Remark 3.** *The specific form of the optimal classifier depends on the sign of $c_1 - d_1 \mathcal{L}^*$, and $\mathcal{L}^*$ is often unknown. In practice, one can often estimate loose upper and lower bounds of $\mathcal{L}^*$ to determine the classifier.*

A number of useful results can be evaluated directly as instances of Theorem 2. For the $F_\beta$ measure, we have that $c_1 = 1 + \beta^2$ and $d_2 = 1$ with all other constants as zero. Thus, $\delta^*_{F_\beta} = \frac{\mathcal{L}^*}{1 + \beta^2}$. This matches the optimal threshold for $F_1$ metric specified by Zhao et al. [14]. For precision, we have that $c_1 = 1, d_2 = 1$ and all other constants are zero, so $\delta^*_{\text{Prec}} = \mathcal{L}^*$. This clarifies the observation that in practice, precision can be maximized by predicting only high confidence positives. For true positive rate (recall), we have that $c_1 = 1, d_0 = \pi$ and other constants are zero, so $\delta^*_{\text{TPR}} = 0$ recovering the known result that in practice, recall is maximized by predicting all examples as positives. For the Jaccard similarity coefficient $c_1 = 1, d_1 = -1, d_2 = 1, d_0 = \pi$ and other constants are zero, so $\delta^*_{\text{JAC}} = \frac{\mathcal{L}^*}{1 + \mathcal{L}^*}$.

When $d_1 = d_2 = 0$, the generalized metric is simply a linear combination of the four fundamental quantities. With this form, we can then recover the optimal classifier outlined by Elkan [5] for cost sensitive classification.

**Corollary 4.** *Let $\mathbb{P}$ be a distribution on $\mathcal{X} \times [0, 1]$ that satisfies Assumption 1, and let $\mathcal{L}$ be a performance metric in the family* (3). *Given the constants $\{c_0, c_1, c_2\}$ and $\{d_0, d_1 = 0, d_2 = 0\}$, the optimal threshold* (5) *is $\delta^* = -\frac{c_2}{c_1}$.*

Classification accuracy is in this family, with $c_1 = 2, c_2 = -1$, and it is well-known that $\delta^*_{\text{ACC}} = \frac{1}{2}$. Another case of interest is the AM metric, where $c_1 = 1, c_2 = -\pi$, so $\delta^*_{\text{AM}} = \pi$, as shown in Menon et al. [7].

## 3 Algorithms

The characterization of the Bayes classifier for the family of performance metrics (4) given in Theorem 2 enables the design of practical classification algorithms with strong theoretical properties. In particular, the algorithms that we propose are intuitive and easy to implement. Despite their simplicity, we show that the proposed algorithms are *consistent* with respect to the measure of interest; a desirable property for a classification algorithm. We begin with a description of the algorithms, followed by a detailed analysis of consistency. Let $\{X_i, Y_i\}_{i=1}^n$ denote iid. training instances drawn from a fixed unknown distribution $\mathbb{P}$. For a given $\theta : \mathcal{X} \to \{0, 1\}$, we define the following empirical quantities based on their population analogues: $\text{TP}_n(\theta) = \frac{1}{n} \sum_{i=1}^n \theta(X_i) Y_i$, and $\gamma_n(\theta) = \frac{1}{n} \sum_{i=1}^n \theta(X_i)$. It is clear that $\text{TP}_n(\theta) \xrightarrow{n \to \infty} \text{TP}(\theta; \mathbb{P})$ and $\gamma_n(\theta) \xrightarrow{n \to \infty} \gamma(\theta; \mathbb{P})$.

Consider the empirical measure:

$$\mathcal{L}_n(\theta) = \frac{c_1 \mathrm{TP}_n(\theta) + c_2 \gamma_n(\theta) + c_0}{d_1 \mathrm{TP}_n(\theta) + d_2 \gamma_n(\theta) + d_0}, \tag{6}$$

corresponding to the population measure $\mathcal{L}(\theta; \mathbb{P})$ in (4). It is expected that $\mathcal{L}_n(\theta)$ will be close to the $\mathcal{L}(\theta; \mathbb{P})$ when the sample is sufficiently large (see Proposition 8). For the rest of this manuscript, we assume that $\mathcal{L}^* \leq \frac{c_1}{d_1}$ so $\theta^*(x) = \mathrm{sign}(\eta_x - \delta^*)$. The case where $\mathcal{L}^* > \frac{c_1}{d_1}$ is solved identically.

Our first approach (Two-Step Expected Utility Maximization) is quite intuitive (Algorithm 1): Obtain an estimator $\hat{\eta}_x$ for $\eta_x = \mathbb{P}(Y = 1|x)$ by performing ERM on the sample using a proper loss function [11]. Then, maximize $\mathcal{L}_n$ defined in (6) with respect to the threshold $\delta \in (0, 1)$. The optimization required in the third step is one dimensional, thus a global minimizer can be computed efficiently in many cases [18]. In experiments, we use (regularized) logistic regression on a training sample to obtain $\hat{\eta}$.

---

**Algorithm 1:** Two-Step EUM

**Input**: Training examples $\mathcal{S} = \{X_i, Y_i\}_{i=1}^n$ and the utility measure $\mathcal{L}$.
1. Split the training data $\mathcal{S}$ into two sets $\mathcal{S}_1$ and $\mathcal{S}_2$.
2. Estimate $\hat{\eta}_x$ using $\mathcal{S}_1$, define $\hat{\theta}_\delta = \mathrm{sign}(\hat{\eta}_x - \delta)$
3. Compute $\hat{\delta} = \arg\max_{\delta \in (0,1)} \mathcal{L}_n(\hat{\theta}_\delta)$ on $\mathcal{S}_2$.

**Return**: $\hat{\theta}_{\hat{\delta}}$

---

Our second approach (Weighted Empirical Risk Minimization) is based on the observation that empirical risk minimization (ERM) with suitably weighted loss functions yields a classifier that thresholds $\eta_x$ appropriately (Algorithm 2). Given a convex surrogate $\ell(t, y)$ of the 0-1 loss, where $t$ is a real-valued prediction and $y \in \{0, 1\}$, the $\delta$-weighted loss is given by [9]:

$$\ell_\delta(t, y) = (1 - \delta)1_{\{y=1\}}\ell(t, 1) + \delta 1_{\{y=0\}}\ell(t, 0).$$

Denote the set of real valued functions as $\Phi$; we then define $\hat{\theta}_\delta$ as:

$$\hat{\phi}_\delta = \arg\min_{\phi \in \Phi} \frac{1}{n} \sum_{i=1}^n \ell_\delta(\phi(X_i), Y_i) \tag{7}$$

then set $\hat{\theta}_\delta(x) = \mathrm{sign}(\hat{\phi}_\delta(x))$. Scott [9] showed that such an estimated $\hat{\theta}_\delta$ is consistent with $\theta_\delta = \mathrm{sign}(\eta_x - \delta)$. With the classifier defined, maximize $\mathcal{L}_n$ defined in (6) with respect to the threshold $\delta \in (0, 1)$.

---

**Algorithm 2:** Weighted ERM

**Input**: Training examples $\mathcal{S} = \{X_i, Y_i\}_{i=1}^n$, and the utility measure $\mathcal{L}$.
1. Split the training data $\mathcal{S}$ into two sets $\mathcal{S}_1$ and $\mathcal{S}_2$.
2. Compute $\hat{\delta} = \arg\max_{\delta \in (0,1)} \mathcal{L}_n(\hat{\theta}_\delta)$ on $\mathcal{S}_2$.
**Sub-algorithm:** Define $\hat{\theta}_\delta(x) = \mathrm{sign}(\hat{\phi}_\delta(x))$ where $\hat{\phi}_\delta(x)$ is computed using (7) on $\mathcal{S}_1$.

**Return**: $\hat{\theta}_{\hat{\delta}}$

---

**Remark 5.** *When $d_1 = d_2 = 0$, the optimal threshold does not depend on $\mathcal{L}^*$ (Corollary 4). We may then employ simple sample-based plugin estimates $\hat{\delta}_S$.*

A benefit of using such plugin estimates is that the classification algorithms can be simplified while maintaining consistency. Given such a sample-based plugin estimate $\hat{\delta}_S$, Algorithm 1 then reduces to estimating $\hat{\eta}_x$, and then setting $\hat{\theta}_{\hat{\delta}_S} = \mathrm{sign}(\hat{\eta}_x - \hat{\delta}_S)$, Algorithm 2 reduces to a single ERM (7) to estimate $\hat{\phi}_{\hat{\delta}_S}(x)$, and then setting $\hat{\theta}_{\hat{\delta}_S}(x) = \mathrm{sign}(\hat{\phi}_{\hat{\delta}_S}(x))$. In the case of AM measure, the threshold is given by $\delta^* = \pi$. A consistent estimator for $\pi$ is all that is required (see [7]).

## 3.1 Consistency of the proposed algorithms

An algorithm is said to be $\mathcal{L}$-consistent if the learned classifier $\hat{\theta}$ satisfies $\mathcal{L}^* - \mathcal{L}(\hat{\theta}) \xrightarrow{P} 0$ i.e., for every $\epsilon > 0$, $\mathbb{P}(|\mathcal{L}^* - \mathcal{L}(\hat{\theta})| < \epsilon) \to 1$, as $n \to \infty$.

We begin the analysis from the simplest case when $\delta^*$ is independent of $\mathcal{L}^*$ (Corollary 4). The following proposition, which generalizes Lemma 1 of [7], shows that maximizing $\mathcal{L}$ is equivalent to minimizing $\delta^*$-weighted risk. As a consequence, it suffices to minimize a suitable surrogate loss $\ell_{\delta^*}$ on the training data to guarantee $\mathcal{L}$-consistency.

**Proposition 6.** *Assume $\delta^* \in (0,1)$ and $\delta^*$ is independent of $\mathcal{L}^*$, but may depend on the distribution $\mathbb{P}$. Define $\delta^*$-weighted risk of a classifier $\theta$ as*

$$R_{\delta^*}(\theta) = E_{(x,y)\sim\mathbb{P}}\big[(1-\delta^*)1_{\{y=1\}}1_{\{\theta(x)=0\}} + \delta^*1_{\{y=0\}}1_{\{\theta(x)=1\}}\big],$$

$$\text{then,} \quad R_{\delta^*}(\theta) - \min_{\theta} R_{\delta^*}(\theta) = \frac{1}{c_1}(\mathcal{L}^* - \mathcal{L}(\theta)).$$

The proof is simple, and we defer it to Appendix B. Note that the key consequence of Proposition 6 is that if we know $\delta^*$, then simply optimizing a weighted surrogate loss as detailed in the proposition suffices to obtain a consistent classifier. In the more practical setting where $\delta^*$ is not known exactly, we can then compute a sample based estimate $\hat{\delta}_S$. We briefly mentioned in the previous section how the proposed Algorithms 1 and 2 simplify in this case. Using the plug-in estimate $\hat{\delta}_S$ such that $\hat{\delta}_S \xrightarrow{p} \delta^*$ in the algorithms directly guarantees consistency, under mild assumptions on $\mathbb{P}$ (see Appendix A for details). The proof for this setting essentially follows the arguments in [7], given Proposition 6.

Now, we turn to the general case, i.e. when $\mathcal{L}$ is an arbitrary measure in the class (4) such that $\delta^*$ is difficult to estimate directly. In this case, both the proposed algorithms estimate $\delta$ to optimize the empirical measure $\mathcal{L}_n$. We employ the following proposition which establishes bounds on $\mathcal{L}$.

**Proposition 7.** *Let the constants $a_{ij}, b_{ij}$ for $i,j \in \{0,1\}$, $a_0$, and $b_0$ be non-negative and, without loss of generality, take values from $[0,1]$. Then, we have:*

1. *$-2 \le c_1, d_1 \le 2, -1 \le c_2, d_2 \le 1$, and $0 \le c_0, d_0 \le 2(1+\pi)$.*

2. *$\mathcal{L}$ is bounded, i.e. for any $\theta$, $0 \le \mathcal{L}(\theta) \le L := \frac{a_0 + \max_{i,j \in \{0,1\}} a_{ij}}{b_0 + \min_{ij \in \{0,1\}} b_{ij}}$.*

The proofs of the main results in Theorem 10 and 11 rely on the following Lemmas 8 and 9 on how the empirical measure converges to the population measure at a steady rate. We defer the proofs to Appendix B.

**Lemma 8.** *For any $\epsilon > 0$, $\lim_{n \to \infty} \mathbb{P}(|\mathcal{L}_n(\theta) - \mathcal{L}(\theta)| < \epsilon) = 1$. Furthermore, with probability at least $1 - \rho$, $|\mathcal{L}_n(\theta) - \mathcal{L}(\theta)| < \frac{(C+LD)r(n,\rho)}{B - Dr(n,\rho)}$, where $r(n,\rho) = \sqrt{\frac{1}{2n}\ln\frac{4}{\rho}}$, $L$ is an upper bound on $\mathcal{L}(\theta)$, $B \ge 0, C \ge 0, D \ge 0$ are constants that depend on $\mathcal{L}$ (i.e. $c_0, c_1, c_2, d_0, d_1$ and $d_2$).*

Now, we show a uniform convergence result for $\mathcal{L}_n$ with respect to maximization over the threshold $\delta \in (0,1)$.

**Lemma 9.** *Consider the function class of all thresholded decisions $\Theta = \{1_{\{\phi(x) > \delta\}} \, \forall \delta \in (0,1)\}$ for a $[0,1]$-valued function $\phi : \mathcal{X} \to [0,1]$. Define $\tilde{r}(n,\rho) = \sqrt{\frac{32}{n}\big[\ln(en) + \ln\frac{16}{\rho}\big]}$. If $\tilde{r}(n,\rho) < \frac{B}{D}$ (where $B$ and $D$ are defined as in Lemma 8) and $\epsilon = \frac{(C+LD)\tilde{r}(n,\rho)}{B - D\tilde{r}(n,\rho)}$, then with prob. at least $1 - \rho$,*

$$\sup_{\theta \in \Theta} |\mathcal{L}_n(\theta) - \mathcal{L}(\theta)| < \epsilon.$$

We are now ready to state our main results concerning the consistency of the two proposed algorithms.

**Theorem 10.** *(Main Result 2) If the estimate $\hat{\eta}_x$ satisfies $\hat{\eta}_x \xrightarrow{p} \eta_x$, Algorithm 1 is $\mathcal{L}$-consistent.*

Note that we can obtain an estimate $\hat{\eta}_x$ with the guarantee that $\hat{\eta}_x \xrightarrow{p} \eta_x$ by using a strongly proper loss function [19] (e.g. logistic loss) (see Appendix B).

**Theorem 11.** *(Main Result 3) Let $\ell : \mathbb{R} : [0, \infty)$ be a* classification-calibrated *convex (margin) loss (i.e. $\ell'(0) < 0$) and let $\ell_\delta$ be the corresponding weighted loss for a given $\delta$ used in the weighted ERM* (7). *Then, Algorithm 2 is $\mathcal{L}$-consistent.*

Note that loss functions used in practice such as hinge and logistic are *classification-calibrated* [8].

## 4 Experiments

We present experiments on synthetic data where we observe that measures from our family indeed are maximized by thresholding $\eta_x$. We also compare the two proposed algorithms on benchmark datasets on two specific measures from the family.

### 4.1 Synthetic data: Optimal decisions

We evaluate the Bayes optimal classifiers for common performance metrics to empirically verify the results of Theorem 2. We fix a domain $\mathcal{X} = \{1, 2, \ldots 10\}$, then we set $\mu(x)$ by drawing random values uniformly in $(0, 1)$, and then normalizing these. We set the conditional probability using a sigmoid function as $\eta_x = \frac{1}{1+\exp(-wx)}$, where $w$ is a random value drawn from a standard Gaussian. As the optimal threshold depends on the Bayes risk $\mathcal{L}^*$, the Bayes classifier cannot be evaluated using plug-in estimates. Instead, the Bayes classifier $\theta^*$ was obtained using an exhaustive search over all $2^{10}$ possible classifiers. The results are presented in Fig. 1. For different metrics, we plot $\eta_x$, the predicted optimal threshold $\delta^*$ (which depends on $\mathbb{P}$) and the Bayes classifier $\theta^*$. The results can be seen to be consistent with Theorem 2 i.e. the (exhaustively computed) Bayes optimal classifier matches the thresholded classifier detailed in the theorem.

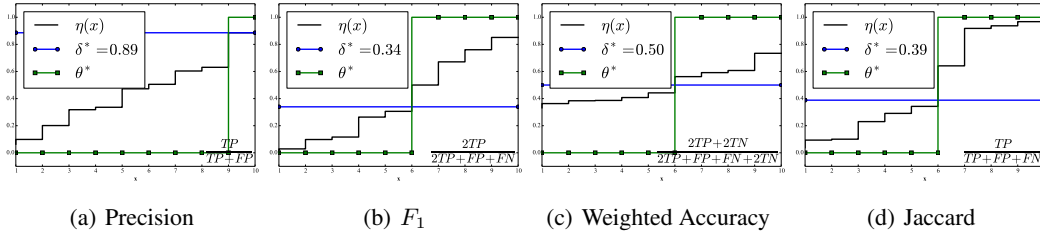

| (a) Precision | (b) $F_1$ | (c) Weighted Accuracy | (d) Jaccard |

Figure 1: Simulated results showing $\eta_x$, optimal threshold $\delta^*$ and Bayes classifier $\theta^*$.

### 4.2 Benchmark data: Performance of the proposed algorithms

We evaluate the two algorithms on several benchmark datasets for classification. We consider two measures, $F_1$ defined as in Section 2 and Weighted Accuracy defined as $\frac{2(TP+TN)}{2(TP+TN)+FP+FN}$. We split the training data $\mathcal{S}$ into two sets $\mathcal{S}_1$ and $\mathcal{S}_2$: $\mathcal{S}_1$ is used for estimating $\hat{\eta}_x$ and $\mathcal{S}_2$ for selecting $\delta$. For Algorithm 1, we use logistic loss on the samples (with $L_2$ regularization) to obtain estimate $\hat{\eta}_x$. Once we have the estimate, we use the model to obtain $\hat{\eta}_x$ for $x \in \mathcal{S}_2$, and then use the values $\hat{\eta}_x$ as candidate $\delta$ choices to select the optimal threshold (note that the empirical best lies in the choices). Similarly, for Algorithm 2, we use a weighted logistic regression, where the weights depend on the threshold as detailed in our algorithm description. Here, we grid the space $[0, 1]$ to find the best threshold on $\mathcal{S}_2$. Notice that this step is embarrassingly parallelizable. The granularity of the grid depends primarily on class imbalance in the data, and varies with datasets. We also compare the two algorithms with the standard empirical risk minimization (ERM) - regularized logistic regression with threshold $1/2$.

First, we optimize for the $F_1$ measure on four benchmark datasets: (1) REUTERS, consisting of news 8293 articles categorized into 65 topics (obtained the processed dataset from [20]). For each topic, we obtain a highly imbalanced binary classification dataset with the topic as the positive class and the rest as negative. We report the average $F_1$ measure over all the topics (also known as macro-$F_1$ score). Following the analysis in [6], we present results for averaging over topics that had at least $C$ positives in the training (5946 articles) as well as the test (2347 articles) data. (2) LETTERS dataset consisting of 20000 handwritten letters (16000 training and 4000 test instances)

from the English alphabet (26 classes, with each class consisting of at least 100 positive training instances). (3) SCENE dataset (UCI benchmark) consisting of 2230 images (1137 training and 1093 test instances) categorized into 6 scene types (with each class consisting of at least 100 positive instances). (4) WEBPAGE binary text categorization dataset obtained from [21], consisting of 34780 web pages (6956 train and 27824 test), with only about 182 positive instances in the train. All the datasets, except SCENE, have a high class imbalance. We use our algorithms to optimize for the $F_1$ measure on these datasets. The results are presented in Table 1. We see that both algorithms perform similarly in many cases. A noticeable exception is the SCENE dataset, where Algorithm 1 is better by a large margin. In REUTERS dataset, we observe that as the number of positive instances $C$ in the training data increases, the methods perform significantly better, and our results align with those in [6] on this dataset. We also find, albeit surprisingly, that using a threshold $1/2$ performs competitively on this dataset.

| DATASET | C | ERM | Algorithm 1 | Algorithm 2 |
|---|---|---|---|---|
|  | 1 | 0.5151 | 0.4980 | 0.4855 |
| REUTERS | 10 | 0.7624 | 0.7600 | 0.7449 |
| (65 classes) | 50 | 0.8428 | 0.8510 | 0.8560 |
|  | 100 | 0.9675 | 0.9670 | 0.9670 |
| LETTERS (26 classes) | 1 | 0.4827 | 0.5742 | 0.5686 |
| SCENE (6 classes) | 1 | 0.3953 | 0.6891 | 0.5916 |
| WEB PAGE (binary) | 1 | 0.6254 | 0.6269 | 0.6267 |

Table 1: Comparison of methods: F1 measure. First three are multi-class datasets: F1 is computed individually for each class that has at least $C$ positive instances (in both the train and the test sets) and then averaged over classes (macro-F1).

Next we optimize for the Weighted Accuracy measure on datasets with less class imbalance. In this case, we can see that $\delta^* = 1/2$ from Theorem 2. We use four benchmark datasets: SCENE (same as earlier), IMAGE (2068 images: 1300 train, 1010 test) [22], BREAST CANCER (683 instances: 463 train, 220 test) and SPAMBASE (4601 instances: 3071 train, 1530 test) [23]. Note that the last three are binary datasets. The results are presented in Table 2. Here, we observe that all the methods perform similarly, which conforms to our theoretical guarantees of consistency.

| DATASET | ERM | Algorithm 1 | Algorithm 2 |
|---|---|---|---|
| SCENE | 0.9000 | 0.9000 | 0.9105 |
| IMAGE | 0.9060 | 0.9063 | 0.9025 |
| BREAST CANCER | 0.9860 | 0.9910 | 0.9910 |
| SPAMBASE | 0.9463 | 0.9550 | 0.9430 |

Table 2: Comparison of methods: Weighted Accuracy defined as $\frac{2(TP+TN)}{2(TP+TN)+FP+FN}$. Here, $\delta^* = 1/2$. We observe that the two algorithms are consistent (ERM thresholds at $1/2$).

## 5 Conclusions and Future Work

Despite the importance of binary classification, theoretical results identifying optimal classifiers and consistent algorithms for many performance metrics used in practice remain as open questions. Our goal in this paper is to begin to answer these questions. We have considered a large family of generalized performance measures that includes many measures used in practice. Our analysis shows that the optimal classifiers for such measures can be characterized as the sign of the thresholded conditional probability of the positive class, with a threshold that depends on the specific metric. This result unifies and generalizes known special cases. We have proposed two algorithms for consistent estimation of the optimal classifiers. While the results presented are an important first step, many open questions remain. It would be interesting to characterize the convergence rates of $\mathcal{L}(\hat{\theta}) \xrightarrow{p} \mathcal{L}(\theta^*)$ as $\hat{\theta} \xrightarrow{p} \theta^*$, using surrogate losses similar in spirit to how excess 0-1 risk is controlled through excess surrogate risk in [8]. Another important direction is to characterize the entire family of measures for which the optimal is given by thresholded $P(Y = 1|x)$. We would like to extend our analysis to the multi-class and multi-label domains as well.

**Acknowledgments:** This research was supported by NSF grant CCF-1117055 and NSF grant CCF-1320746. P.R. acknowledges the support of ARO via W911NF-12-1-0390 and NSF via IIS-1149803, IIS-1320894.

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
