[Supplementary Material · cameraready_supp.pdf]

## Appendix A

**Lemma 12.** *Let $\mathcal{F} = \{f : \mathcal{X} \mapsto \mathfrak{R}\}$, the constraint set $\mathcal{C} \subset \mathcal{F}$, and the functional $\mathcal{G} : \mathcal{F} \mapsto \mathfrak{R}$, consider the optimization problem:*

$$f^* = \arg\max_{f \in \mathcal{F}} \mathcal{G}(f) \quad s.t. \quad f \in \mathcal{C}$$

*If the Fréchet derivative $\nabla \mathcal{G}(f)$ exists, then $f^*$ is locally optimal iff. $f^* \in \mathcal{C}$ and:*

$$\langle \nabla \mathcal{G}(f^*), f^* - f \rangle \geq 0 \quad \forall \, f \in \mathcal{C},$$

$$\implies \int_{x \in \mathcal{X}} [\nabla \mathcal{G}(f^*)]_x f^*(x) dx \geq \int_{x \in \mathcal{X}} [\nabla \mathcal{G}(f^*)]_x f(x) dx \quad \forall \, f \in \mathcal{C}.$$

Lemma 12 is a generalization of the well known first order condition for optimality of finite dimensional optimization problems [24, Section 4.2.3] to optimization of smooth functionals.

**Proposition 13.** *Let $\mathcal{L}$ be a measure of the form (4), and $\hat{\delta}_S$ be some estimator of its optimal threshold $\delta^*$. Assume $\hat{\delta}_S \in (0, 1)$ and $\hat{\delta}_S \xrightarrow{p} \delta^*$. Also assume the cumulative distribution of $\eta_x$ conditioned on $Y = 1$ and on $Y = 0$, $F_{\eta_x|Y=1}(z) = \mathbb{P}(\eta_x \leq z|Y = 1)$ and $F_{\eta_x|Y=0}(z) = \mathbb{P}(\eta_x \leq z|Y = 0)$ are continuous at $z = \delta^*$. Let the classifier be given by one of the following:*

(a) *the classifier $\hat{\theta}_{\hat{\delta}_S}(x) = \text{sign}(\hat{\eta}_x - \hat{\delta}_S)$, where $\hat{\eta}$ is a class probability estimate that satisfies $E_x[|\hat{\eta}_x - \eta_x|^r] \xrightarrow{p} 0$ for some $r \geq 1$,*

(b) *the classifier $\hat{\theta}_{\hat{\delta}_S} = \text{sign}(\hat{\phi}_{\hat{\delta}_S})$, the empirical minimizer of the ERM (7) using a suitably calibrated convex loss $\ell_{\hat{\delta}_S}$ [9],*

*then $\hat{\theta}_{\hat{\delta}_S}$ is $\mathcal{L}$-consistent.*

*Proof.* Given Proposition 6, the proofs for parts (a) and (b) essentially follow from the arguments in [7] for consistency with respect to the AM measure. Under the stated assumptions, the decomposition Lemma (Lemma 2) of [7] holds: For a classifier $\hat{\theta}$, if

$$R_{\hat{\delta}_S}(\hat{\theta}) - \min_{\theta} R_{\hat{\delta}_S}(\theta) \xrightarrow{p} 0 \quad \text{then,} \quad \mathcal{L}^* - \mathcal{L}(\hat{\theta}) \xrightarrow{p} 0$$

This allows us to directly invoke Theorems 5 and Theorems 6 of [7] giving us the desired $\mathcal{L}$-consistency in parts (a) and (b) respectively. $\qquad\square$

## Appendix B: Proofs

### Proof of Theorem 2

*Proof.* Let $\mathcal{F} = \{f : \mathcal{X} \mapsto \mathfrak{R}\}$, and note that $\Theta \subset \mathcal{F}$. We consider a continuous extension of (4) by extending the domain of $\mathcal{L}$ from $\Theta$ to $\mathcal{F}$. This results in the following optimization:

$$f^* = \arg\max_{f \in \mathcal{F}} \mathcal{L}(f) \quad s.t. \quad f \in \Theta \tag{8}$$

It is clear that (4) is equivalent to (8), and the minima coincide i.e. $f^* = \theta^*$. The Fréchet derivative of $\mathcal{L}$ evaluated at $x$ is given by:

$$[\nabla \mathcal{L}(f)]_x = \frac{1}{(c_1 - d_1 L(f)) D_r(f)} \left[ \eta_x - \frac{d_2 \mathcal{L}(f) - c_2}{c_1 - d_1 \mathcal{L}(f)} \right] \mu(x)$$

where $D_r(f)$ is denominator of $\mathcal{L}(f)$. A function $f^* \in \Theta$ optimizes $\mathcal{L}$ if $f^* \in \Theta$ and (Lemma 12):

$$\int_{x \in \mathcal{X}} [\nabla \mathcal{L}(f^*)]_x f(x) dx \geq \int_{x \in \mathcal{X}} [\nabla \mathcal{L}(f^*)]_x f^*(x) dx \quad \forall \, f \in \Theta.$$

Thus, when $c_1 \geq d_1 \mathcal{L}^*$, a necessary condition for local optimality is that the sign of $f^*$ and the sign of $[\nabla \mathcal{L}(f^*)]$ agree pointwise wrt. $x$. This is equivalent to the condition that $\text{sign}(f^*) = \text{sign}(\eta_x - \delta^*)$. Combining this result with the constraint set $f \in \Theta$, we have that $f^* = \text{sign}(f^*)$, thus $f^* = \text{sign}(\eta_x - \delta^*)$ is locally optimal. Finally, we note that $f^* = \text{sign}(\eta_x - \delta^*)$ is unique for $f \in \Theta$, thus $f^*$ is globally optimal. The proof for $c_1 < d_1 \mathcal{L}^*$ follows using similar arguments. $\quad\square$

**Proof of Proposition 6**

*Proof.* From Corollary 4 we know $\delta^* = -\frac{c_2}{c_1}$. Since $0 < \delta^* < 1$, and $c_1 < 1$ from Proposition 7, we have $1 > c_1 > 0$. We can rewrite $\mathcal{L}(\theta)$ as $\mathcal{L}(\theta) = c_1[(1-\delta^*)\text{TP} + \delta^*\text{TN}] + \tilde{A}$, where $\tilde{A}$ is a constant. We have:

$$
\begin{aligned}
R_{\delta^*}(\theta) &= E_{(x,y)\sim\mathbb{P}}\left[\left((1-\delta^*)1_{\{y=1\}} + \delta^* 1_{\{y=0\}}\right).1_{\{\theta(x)\neq y\}}\right]\\
&= (1-\delta^*)P(y=1,\theta(x)=0) + \delta^* P(y=0,\theta(x)=1)\\
&= (1-\delta^*)\text{FN} + \delta^*\text{FP}\\
&= (1-\delta^*)(\pi - \text{TP}) + \delta^*(1 - \pi - \text{TN})\\
&= (1-\delta^*)\pi + \delta^*(1-\pi) - \left((1-\delta^*)\text{TP} + \delta^*\text{TN}\right)\\
&= (1-\delta^*)\pi + \delta^*(1-\pi) + \frac{\tilde{A}}{c_1} - \frac{1}{c_1}\mathcal{L}(\theta).
\end{aligned}
$$

Observing that $(1-\delta^*)\pi + \delta^*(1-\pi) + \frac{\tilde{A}}{c_1}$ is a constant independent of $\theta$, the proof is complete. $\square$

**Proof of Lemma 8**

*Proof.* For a given $\theta, \epsilon_1 > 0, \rho > 0$, there exists an $N$ such that for any $n > N$, $\mathbb{P}(|\text{TP}_n(\theta) - \text{TP}(\theta)| < \epsilon_1) > 1 - \rho/2$ and $\mathbb{P}(|\gamma_n(\theta) - \gamma(\theta)| < \epsilon_1) > 1 - \rho/2$. By union bound, the two events simultaneously hold with probability at least $1 - \rho$. Let $\tilde{c}_1 = 1/|c_1|$ if $c_1 \neq 0$ else $\tilde{c}_1 = 0$. Define $\tilde{c}_2, \tilde{d}_1, \tilde{d}_2$ similarly. Now define $C = \max(\tilde{c}_1, \tilde{c}_2)$ and $D = \max(\tilde{d}_1, \tilde{d}_2)$. Observe that either $C > 0$ or $D > 0$ otherwise $\mathcal{L}$ is a constant. Now for a given $\epsilon > 0$, after some simple algebra, we need

$$\epsilon_1 \leq \frac{(d_1\text{TP}(\theta) + d_2\gamma(\theta) + d_0)\epsilon}{D(\mathcal{L}(\theta) + \epsilon) + C}.$$

Choosing some $\epsilon_1$ satisfying the upper bound above guarantees $\mathcal{L}(\theta) - \epsilon < \mathcal{L}_n(\theta) < \mathcal{L}(\theta) + \epsilon$. Thus for all $n > N$ implied by this $\epsilon_1$ and $\rho$, $P(|\mathcal{L}_n(\theta) - \mathcal{L}(\theta)| < \epsilon) > 1 - \rho$ holds.

Now, for the rate of convergence, Hoeffding's inequality with $\rho = 4e^{-2n\epsilon_1^2}$ (or $\epsilon_1 = \sqrt{\frac{1}{2n}\ln\frac{4}{\rho}}$) gives us $\mathbb{P}(|\text{TP}_n(\theta) - \text{TP}(\theta)| < \epsilon_1) > 1 - \rho/2$ and $\mathbb{P}(|\gamma_n(\theta) - \gamma(\theta)| < \epsilon_1) > 1 - \rho/2$. Choose $\epsilon_1 > 0$ as a function of $\epsilon$ such that it is sufficiently small, i.e. $\epsilon_1 \leq \frac{(d_1\text{TP}(\theta) + d_2\gamma(\theta) + d_0)\epsilon}{D(\mathcal{L}(\theta) + \epsilon) + C}$. We know $\mathcal{L}(\theta) \leq L$ for any $\theta$ (from Proposition 7), therefore $D(\mathcal{L}(\theta) + \epsilon) + C < D(L + \epsilon) + C$. Furthermore, $d_1\text{TP}(\theta) + d_2\gamma(\theta) + d_0 > b_0 + \min(b_{00}, b_{11}, b_{01}, b_{10}) := B$. We can choose $\epsilon_1 = \frac{B\epsilon}{D(L+\epsilon)+C} \leq \frac{(d_1\text{TP}(\theta)+d_2\gamma(\theta)+d_0)\epsilon}{D(\mathcal{L}(\theta)+\epsilon)+C}$ or $\epsilon = \frac{(C+LD)\epsilon_1}{B-D\epsilon_1}$. From the first part of the lemma, we know $P(|\mathcal{L}_n(\theta) - \mathcal{L}(\theta)| < \epsilon) > 1 - \rho$ holds with probability at least $\rho$. This completes the proof. $\square$

**Proof of Lemma 9**

*Proof.* Let $\rho = 16e^{\ln(en)-n\epsilon_1^2/32}$, then $\epsilon_1 = \tilde{r}(n, \rho)$. Using Lemma 29.1 in [25], we obtain:

$$\mathbb{P}\left[\sup_{\theta\in\Theta}|\text{TP}_n(\theta) - \text{TP}(\theta)| < \epsilon_1\right] > 1 - \rho/2\,.$$

By union bound, the inequalities $\mathbb{P}\left[\sup_{\theta\in\Theta}|\text{TP}_n(\theta)-\text{TP}(\theta)| < \epsilon_1\right]$ and $\mathbb{P}\left[\sup_{\theta\in\Theta}|\gamma_n(\theta)-\gamma(\theta)| < \epsilon_1\right]$ simultaneously hold with probability at least $1 - \rho$. If $n$ is large enough that $\tilde{r}(n, \rho) < \frac{B}{D}$, then from Proposition 8 we know that, for any given $\theta$, $|\mathcal{L}_n(\theta) - \mathcal{L}(\theta)| < \frac{(C+LD)\tilde{r}(n,\rho)}{B-D\tilde{r}(n,\rho)}$ with probability at least $1 - \rho$. The lemma follows. $\square$

**Proof of Theorem 10**

*Proof.* Using a strongly proper loss function [19] and its corresponding link function $\psi$, and an appropriate function class to minimize the empirical loss, we can obtain a class probability estimator $\hat{\eta}$ such that $E_x\left[|\hat{\eta}_x - \eta_x|^2\right] \to 0$ (from Theorem 5 in [26]). Convergence in mean implies convergence

in probability and so we have $\hat{\eta} \xrightarrow{p} \eta$. Now let $\theta_\delta^* = \text{sign}(\eta_x - \delta)$. Recall that $\hat{\delta}$ denotes the empirical maximizer obtained in Step 3. Now, since $\mathcal{L}_n(\theta_{\hat{\delta}}^*) \geq \mathcal{L}_n(\theta_{\delta^*}^*)$, it follows that:

$$
\begin{aligned}
\mathcal{L}^* - \mathcal{L}(\theta_{\hat{\delta}}^*) &= \mathcal{L}^* - \mathcal{L}_n(\theta_{\hat{\delta}}^*) + \mathcal{L}_n(\theta_{\hat{\delta}}^*) - \mathcal{L}(\theta_{\hat{\delta}}^*) \\
&\leq \mathcal{L}^* - \mathcal{L}_n(\theta_{\delta^*}^*) + \mathcal{L}_n(\theta_{\hat{\delta}}^*) - \mathcal{L}(\theta_{\hat{\delta}}^*) \\
&\leq 2 \sup_\delta |\mathcal{L}(\theta_\delta^*) - \mathcal{L}_n(\theta_\delta^*)| \\
&\leq 2\epsilon \xrightarrow{p} 0
\end{aligned}
$$

where $\epsilon$ is defined as in Lemma 9. The last step is true by instantiating Lemma 9 with the thresholded classifiers corresponding to $\phi(x) = \eta_x$. $\qquad \square$

**Proof of Theorem 11**

*Proof.* For a fixed $\delta$, $E_{(X,Y) \sim \mathbb{P}}[\ell_\delta(\hat{\theta}_\delta(X), Y)] \to \min_\theta E_{(X,Y) \sim \mathbb{P}}[\ell_\delta(\theta(X), Y)]$. With the understanding that the surrogate loss $\ell_\delta$ (i.e. the $\ell_\delta$-risk) satisfies regularity assumptions and the minimizer is unique, the weighted empirical risk minimizer also converges to the corresponding Bayes classifier [9]; i.e., we have $\hat{\theta}_\delta \xrightarrow{p} \theta_\delta^*$. In particular, $\hat{\theta}_{\hat{\delta}} \xrightarrow{p} \theta_{\hat{\delta}}^* = \text{sign}(\eta_x - \hat{\delta})$. Let $\hat{\delta}$ denote the empirical maximizer obtained in Step 2. Now, by using an argument identical to the one in Theorem 10 we can show that $\mathcal{L}^* - \mathcal{L}(\theta_{\hat{\delta}}^*) \leq 2\epsilon \xrightarrow{p} 0$. $\qquad \square$