[Reviews · NeurIPS 2014]

Submitted by Assigned_Reviewer_20

The paper provides a study and accompanying algorithms for binary problems where performance metrics go beyond the mere classification accuracy; such performances are F-measure, AM-measure, Jaccard similarity coefficient, and, of course, precision and recall measures. The findings of the authors are a few:
a) they show that there are many performance measures that can be expressed as ratio of linear combinations of the false positive (FP) rates, true positive rates (TP), true negatives (TN) and false negative (FN) rates. Working out a bit these ratio, it turns out that they can be expressed with respect to TP and probability for the classifier to predict a +1 class.

b) a consequence of a) is that it is possible to show the Bayes classifier (which optimizes the performance metric at hand) is simply of the form sign(eta_x - \delta) where eta_x is the conditional probability for Y=1 given x.

c) 2 consistent learning algorithms building upon b) (and therefore a)) are provided and proof of consistency (of one of them) are provided (and correct, as far as I have checked)

d) numerical simulations on both controlled and benchmark datasets are provided that illustrate the soundness of the algorithms.

Finally, a few directions for future research such as multiclass classification and multilabel classification are outlined.

Quality
------
The paper is of very good quality, both from the mathematical and writing viewpoints. It makes reference to the pivotal works in the same family of contributions that help understand the proposed contributions. The structure of the paper is very sound and there is a nice balance between the depiction of basic things (cf. the linear ratios), the mathematical part (on consistency), the algorithmic part and the experimental part.

Clarity
------
The paper is very clear. The authors made a good choice in postponing the most technical part to the appendix.

Originality
---------
It is very nice to discover that there is still very interesting (and not just technicall) work to do in the realm of binary classification. This is all the more so that the topic addressed (richer performance measure than the accuracy) are of very high practical interest.

Significance
----------
It is no doubt that this paper will be of very high interest for people working in information retrieval or for those looking for new learning framework to chew from the statistical standpoint.

Minor comments
--------------
- insert a section 2.1 (named, e.g. "Metrics as ratios of linear combinations...") before the current section 2.1, or remove section title 2.1. (If there is only one subsection then there is no need to have any numbering.)
- l. 387: "the methods, the methods"
Summary: This paper addresses the very important topic of learning when performance measures richer than accuracy are taken into account. The authors provide a nice study that shows is is possible to give theoretically relevant and yet simple algorithms to tackle this learning situation, building about simple yet powerful observations as how many performance metrics decompose wrt FP, FN, TP, TN measures.

Submitted by Assigned_Reviewer_40

* Summary
This paper studies the consistency of classification rules that maximizes generalized performance metrics, that are ratios of linear combinations of four populations quantities widely used in practice (FP,TP,FN.TN). The paper shows that the optimal rule for such metrics has the form of sign(P(y|x)-\delta^*), where \delta^* is a threshold that is metric dependent. This unified framework recovers many results for metrics studied in the literature.
Authors devise 2 simple algorithms for estimating p(y|x) and the threshold delta^*. Bayes and statistical consistency of the proposed algorithms is then analyzed. numerical experiments validate the benefits of the proposed algorithm.

*Quality and Clarity:

The paper is very well written and the exposition is clear.
The unified view on those performance metrics, and the equivalence with weighted ERM for fixed threshold is elegant, and the derived theory is very insightful form a practical point view.

Comments :
- In algorithm1 ,the paper is not clear about how to estimate \eta(y|x), it seems to apply an ERM using a surrogate loss function, a better exposition here of the methods could be helpful, it will help digesting theorem 10.

- In algorithm 2, also the loss \ell is also used . emphasizing the relaxation from the 0-1 loss to the surrogate loss \ell might help appreciating theorem 11.

- The surrogate loss used in estimating p(y|x), seems to be coming very late in the analysis. It would be interesting to have the surrogate loss as a starting point in the analysis as it is done in binary classification. What are the difficulties in doing that?

Summary: The paper starts an effort in analyzing consistency of classification rules maximizing generalized performance metrics widely used in practice. It builds a unified theory and a solid framework that the authors and others can build on.

Submitted by Assigned_Reviewer_43

This paper studies optimal classifiers for a family of performance metrics that encompasses a lot of well-known and widely used ones. It essentially shows that within that family, optimal classifiers turn out to be the sign of the thresholded conditional probability of the class. The optimal threshold is derived and is shown to be estimable in a fairly straightforward way. Moreover, two simple algorithms are proposed to provide classifiers which prove to be statistically consistent. Finally, some experiment, led on synthetic or benchmark data, are presented, validating the theoretical results.

The results seem novel, generalize existing ones and are sound. The elements of proof given in the appendix are sufficient. The quality and interest of the results are obvious.

The paper is very nicely written. It makes it very pleasant to understand the problem that is tacked as well as the importance or novelty of the results or how they rely on previous work.

The main originality lies in the definition of this class of performance measures that make it possible to derive a useful generalization of existing specific results. The results themselves, or their proofs are reasonably straightforward, which puts even more emphasis on the relevance of their definition of a class of performance metrics.

Some of the most important perspectives are already mentioned in the conclusion.
However, one may regret there isn't more comments and discussion on the results regarding the consistency.
Some extensions for multi-class or multi-label settings are suggested.
Is there any hope or insight regarding an extension to ranking metrics?
Summary: This paper deals with the study of classification with a fairly general class of performance metrics, addressing the derivation of optimal classification rules, and algorithms being provably consistant with them. It is very clearly written and the results, if not groundbreaking, are novel and open very interesting perspectives.
Author Feedback
Author rebuttal: We thank the reviewers for supportive and helpful comments.

Response to Assigned_Reviewer_20:
We agree with your suggestions for organizing the sections and will modify the manuscript accordingly. You noted that: "2 consistent learning algorithms ... are provided and proof of consistency (of one of them) are provided". We wish to clarify that consistency proofs were provided for both algorithms (Theorem 10 and Theorem 11).

Response to Assigned_Reviewer_40:
* "In Algorithm 1, the paper is not clear about how to estimate \eta(y|x), it seems to apply an ERM using a surrogate loss function, a better exposition here of the methods could be helpful, it will help digesting theorem 10."
As you have noted, \eta(y|x) may be estimated via ERM using any of the large class of proper loss functions proposed in [11] (see lines 232-234). We chose not to include this detail in the algorithm for readability. We will improve the level of detail in the final manuscript as suggested.

* "In Algorithm 2, also the loss \ell is also used, emphasizing the relaxation from the 0­1 loss to the surrogate loss \ell might help appreciating theorem 11."
As noted, Algorithm 2 applies the results of [9], where the weighted 0-1 loss and its surrogates have been analyzed (see line 246 and 256). We will improve the level of detail in the final manuscript as suggested.

* "The surrogate loss used in estimating p(y|x), seems to be coming very late in the analysis. It would be interesting to have the surrogate loss as a starting point in the analysis as it is done in binary classification. What are the difficulties in doing that?"

Our analysis differs from standard manuscripts as we must first determine the Bayes classifier before deriving surrogates. In contrast, the bayes optimal for binary classification with the 0-1 loss is well known, thus most manuscripts need not re-derive it.

Response to Assigned_Reviewer_43:
* "However, one may regret there isn't more comments and discussion on the results regarding the consistency."
Further discussion was omitted due to space constraints and will be included in the extended version of this manuscript.

* "Is there any hope or insight regarding an extension to ranking metrics?"
We agree that the analysis of ranking metrics is indeed a fruitful direction. Thank you for this suggestion. We expect that our approach will be useful, particularly in light of recent results [27]. Ranking metrics will be addressed as part of our future work.

Additional references:
[27] Narasimhan, Harikrishna, and Shivani Agarwal. "On the relationship between binary classification, bipartite ranking, and binary class probability estimation." Advances in Neural Information Processing Systems. 2013.